# Non-Autoregressive Math Word Problem Solver with Unified Tree Structure

**Yi Bin[1], Mengqun Han[2], Wenhao Shi[2], Lei Wang[3], Yang Yang[2]**
**See-Kiong Ng[1], Heng Tao Shen[2]**

[1]National University of Singapore, [2]University of Electronic Science and Technology of China
[3]Singapore Management University

{yi.bin, shenhengtao}@hotmail.com, hanmengqun@foxmail.com

{shiwenhao16, dlyyang}@gmail.com, lei.wang.2019@phdcs.smu.edu.sg, seekiong@nus.edu.sg

## Abstract

Existing MWP solvers employ sequence or binary tree to present the solution expression and decode it from given problem description. However, such structures fail to handle the variants that can be derived via mathematical manipulation, *e.g.*, $(a_1 + a_2) * a_3$ and $a_1 * a_3 + a_2 * a_3$ can both be possible valid solutions for a same problem but formulated as different expression sequences or trees. The multiple solution variants depicting different possible solving procedures for the same input problem would raise two issues: 1) making it hard for the model to learn the mapping function between the input and output spaces effectively, and 2) wrongly indicating *wrong* when evaluating a valid expression variant. To address these issues, we introduce a unified tree structure to present a solution expression, where the elements are permutable and identical for all the expression variants. We propose a novel non-autoregressive solver, named *MWP-NAS*, to parse the problem and deduce the solution expression based on the unified tree. For evaluating the possible expression variants, we design a path-based metric to evaluate the partial accuracy of expressions of a unified tree. The results from extensive experiments conducted on Math23K and MAWPS demonstrate the effectiveness of our proposed MWP-NAS. The codes and checkpoints are available at: https://github.com/mengqunhan/MWP-NAS.

## 1 Introduction

Automatically solving math word problems (MWP) is an important and fundamental task in Artificial Intelligence that has attracted a great deal of research attention throughout the years (Bobrow, 1964; Wang et al., 2017; Zhang et al., 2020; Huang et al., 2018). Figure 1 illustrates an MWP example and its solutions. Given an input word problem description, the MWP solver needs to understand the problem in natural language, figure out the mathematical logic underlying the problem, then for-

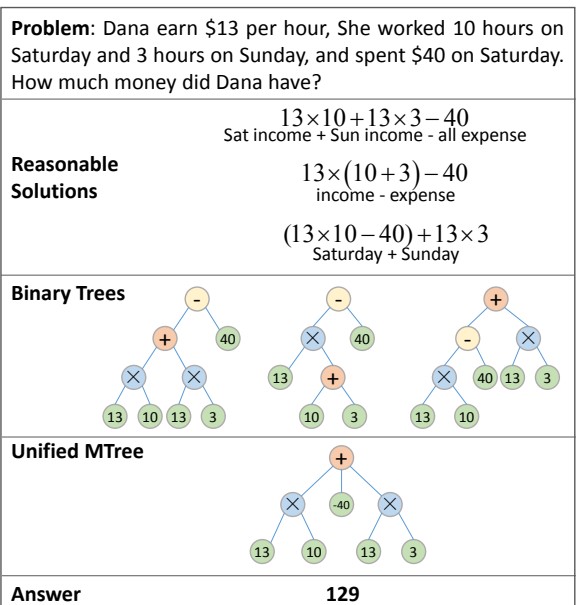

Figure 1: An MWP example with multiple reasonable solutions and corresponding binary trees, but only one unique output based on the unified MTree structure.

mulate the corresponding solution expression by employing suitable mathematical operation symbols (*e.g.*, $+, -, \times, /$) to link the relevant numbers in the problem. Due to the complex reasoning in text and expression, automatically solving math word problems has always been considered as one of the key testbeds for evaluating the intelligence level of an AI agent (Charniak, 1969; Clark, 2015).

The approaches for automatically solving MWP proposed by various researchers over the years can be categorized into three stages: manual features and rule-based at early stage, followed by facilitating quantitative reasoning with semantic parsing, and the use of deep learning in recent years (Zhang et al., 2019). Methods from the first two stages did not work well as they were limited in their capacity for feature representation and language understanding. With the success of deep learning in natural language understanding, a series of neu-

ral solvers for MWP have been proposed recently and achieved superior performance. Inspired by the sequence to sequence modeling in machine translation, several works (Wang et al., 2017; Huang et al., 2018; Wang et al., 2019) considered MWP as a sequence to sequence mapping problem, from natural language to mathematical expression, and sequentially generate the numbers and operators of the expression. However, such sequential representation ignored the arithmetic properties of the expression. As such, the binary tree was introduced to present the expression and achieved tremendous progress with Seq2Tree models (Xie and Sun, 2019; Shen and Jin, 2020; Lin et al., 2021) and Graph2Tree models (Zhang et al., 2020; Li et al., 2020). More recently, the application of pre-trained language models to MWP solving (Liang et al., 2022; Shen et al., 2021) have led to new state-of-the-arts due to the superior capacity and performance in language modelling.

Most existing approaches learn to match exactly the target expression and ignore the alternative ones that can be derived by the arithmetic laws, *e.g.*, commutative law, associative law, and distributive law. As the example shown in Figure 1, we can calculate the answer following either the rules of $income - expense$ or $Saturday + Sunday$ resulting in different expressions and binary trees. However, translating the same problem description to different expression variants is inconsistent with the function mapping, which introduces output indeterminacy and makes the model difficult to optimize. Furthermore, the issue of variant expressions may lead to problems in expression evaluation due to possible false negatives and thus underestimation of the performance of the methods.

To address the above issues, we propose to learn the MWP solver based on a unified form of expression. Specifically, we eliminate the diversity of expression variants and unify them to an identical format with MTree structure (Wang et al., 2022). Unlike having only two branches in binary tree, MTree may contain multiple branches for each node, as shown in Figure 1. Since the expression diversity is mainly introduced by the computational priority, MTree removes the $/$ and $-$ operators, and introduces $\times-$ and $+/$ to make all the operators are with the same superiority and the child nodes of the same parent are permutable. Based on such MTree, Wang et al. (2022) design MTree codes to represent and learn the mapping between prob-

lem and expression. However, such codes break the mathematical relations between numbers (Jie et al., 2022) and may lead to inferior performance. Furthermore, the code dictionary has to be predefined which limits the flexibility and may incur the issue of out-of-vocabulary (OOV) during inference. Motivated by the intuition of goal-driven strategy for tree-structure decoding, we propose a non-autoregressive MWP solver, *MWP-NAS* for short, to integrate with goal-driven decoding based on MTree. Specifically, our MWP-NAS first obtains the semantic representation of problem text via a neural language model, and uses it as the initial goal for MTree decoding. As there exist multiple unordered child nodes for each parent, we employ a non-autoregressive Transformer to explore the comprehensive interactions among the nodes and predict them in parallel. For each sub-goal, we recursively implement such non-autoregressive decoding until all the child nodes are numbers. A cross-goal attention strategy is devised to enable the interactions between goals and make the decomposed sub-goals more accurate. In MTree, as each number could be applied in one of four forms: $\{n, -n, \frac{1}{n}, -\frac{1}{n}\}$, we design a simple classification module to learn the form, which is jointly trained with the non-autoregressive decoder. We also observe that under the current evaluation approaches, the variants of the same expression would be reported as wrong by expression evaluation, *a.k.a* false negative, which may underestimate the ability of the methods. Furthermore, we want to be able to evaluate the ability of a solver not only from the final expression, but also considering the partial correctness of the sub-expressions. Towards this end, we propose two MTree based evaluation metrics: 1) **MTree Acc** measuring the exact matching accuracy of unified MTree, and 2) **MTree IoU** calculating the intersection over union of MTree paths between predicted and ground-truth expressions.

In summary, the main contributions of this paper are as follows:

- We propose a novel non-autoregressive solver for MWP, dubbed MWP-NAS, which implements a goal-driven non-autoregressive manner to decompose the goal to sub-goals based on the unified MTree structure. To enable information passing between goals for sub-goal decomposition, we devise a novel cross-goal attention to make the model leverage information across goals.

- We design two metrics for expression evaluation based on MTree, to better assess the effectiveness of MWP solvers.

- Extensive experiments are conducted on Math23K and MAWPS, and the results demonstrate that our MWP-NAS outperforms all the SoTAs. We also compare several milestone baselines evaluated by our MTree Acc and IoU, and analyze the effectiveness of our MTree-based metrics.

## 2 Related Works

### 2.1 MWP Solving

MWP solving has been attracting wide research attention for a long time (Bobrow, 1964; Bakman, 2007; Kushman et al., 2014; Shi et al., 2015; Mitra and Baral, 2016). Inspired by the superiority of deep neural networks, deep learning based methods have dominated this area and achieved impressive progress. Wang et al. (2017) first regarded MWP as a sequence to sequence mapping problem, and designed an RNN-based seq2seq model, GRU encoder and LSTM decoder in specific, to sequentially generate the numbers and operators of the expression. Based on the seq2seq model, Wang et al. (2018b) and Huang et al. (2018) introduced deep reinforcement learning to MWP solving and achieved promising performance. Robaidek et al. (2018) also made an attempt to employ convolutional neural networks as encoder and decoder. Wang et al. (2019) proposed to predict the expression template and then fill the operator into the template by designing a recursive neural network.

Subsequently, Xie and Sun (2019) proposed a goal-driven expression tree generation strategy, which recursively decomposes the current goal into sub-goals via a top-down manner. The tree-based expression template significantly improved the solution accuracy. Zhang et al. (2020) introduced Quantity Cell Graph and Quantity Comparison Graph to enrich the problem representation, and designed a Graph2Tree framework to learn the expression tree. Cao et al. (2021) applied Direct Acyclic Graph (DAG) to represent the expression and devised a Seq2DAG model, aggregating the numbers and sub-expressions, to obtain the DAG of expression. Jie et al. (2022) started from deductive reasoning and regarded MWP solving as a complex relation extraction problem, then proposed to learn the relation between two quantities iteratively in a bottom-up manner. Bin et al. (2023a) introduced the reexamination process of human and devised a model-agnostic pseudo-dual learning scheme to further improve the performance. With the booming of pre-trained language models (PLM), researchers have been trying to strengthen the models with PLM. Liang et al. (2022) built a number-aware MWP-BERT with PLM to effectively learn the contextual number representation. Using the PLM as encoder, *e.g.*, BERT or RoBERTa, to extract problem text representation also significantly boosts the MWP solving accuracy (Jie et al., 2022; Li et al., 2022; Wang et al., 2022).

### 2.2 Non-Autoregressive Transformer

Non-Autoregressive Transformer (NAT) (Gu et al., 2018) is proposed to accelerate the decoding process in machine translation by generating all the words in parallel. As pointed in (Gu et al., 2018), however, NAT suffers from the multimodality problem and exhibits complete conditional independence, resulting in severe repetition issue in generated texts and inferior generation performance. To address these issues, many works have been proposed to enhance the latent representations by integrating an extra refinement process (Li et al., 2019b; Guo et al., 2020), or design a semi-autoregressive fashion (Mallinson et al., 2022; Wang et al., 2018a). Besides, Huang et al. (2022) introduced a directed acyclic transformer to capture multiple translations and facilitate fast predictions in NAT, demonstrating superior performance. In addition, Bin et al. (2023b) employed NAT for ordering problem which could benefit from the bi-lateral dependencies modelling and avoid the repetition issue by designing an exclusive loss, as well as outfit generation (Ding et al., 2023)

Following (Wang et al., 2022), our work tries to unify the expression using MTree and learn the mapping function between problem text and MTree. Wang et al. (2022) predefined a code dictionary over the training set and may result in OOV and poor generalization. To address these issues, we propose a non-autoregressive decoder to learn the MTree of expression with a goal-driven strategy.

## 3 Methodology

To handle multiple branches in MTree, we propose a non-autoregressive solver, termed as MWP-NAS, for the child nodes generation. As depicted on the left of Figure 2, our MWP-NAS mainly consists

of a Problem Encoder and a Goal-Driven MTree Generator to understand the problem and reason the MTree structure of solution expression, which is used to compute the final answer. To handle multi-branch decoding in MTree generator, we devise a novel non-autoregressive goal decomposer, shown on the right side in Figure 2. All the details will be introduced in the subsequent parts.

## 3.1 Preliminary of MTree

MTree is first introduced to MWP by (Wang et al., 2022), to unify the expression tree structure. Given an expression, we first transform it to a plain expression by removing the brackets with SymPy[1], an arithmetical Python package. The plain expression only contains numbers and four arithmetical operators $\{+, -, \times, /\}$[2], where the operands of $\{+, \times\}$ can be swapped while the ones for $\{-, /\}$ cannot. The unswappable operators also cannot be applied in multi-branch tree because different order of operands would lead to different results. To tackle this issue, two new operators $\{\times-, +/\}$ are introduced to replace $\{-, /\}$. $\times-$ indicates to get the opposite value of the product of multiple numbers, e.g., $\times-$ with operands $\{2, 3\}$ is equal to $-(2 \times 3)$. $+/$ means calculating the reciprocal for the sum of the operands, e.g., $\frac{1}{2+3}$ for operands $\{2, 3\}$. Finally, the operators for MTree are $\{+, \times, \times-, +/\}$, where all of them are able to handle multiple operands and the operands are swappable. The summarized meanings of operators are as follows:

- $+ (\times)$ means the sum (product) of operands, e.g., $1 + 2 + 3$ and $1 \times 2 \times 3$ for the operands $\{1, 2, 3\}$.

- $\times- (+/)$ means the opposite (reciprocal) of the productive (sum) of the operands, e.g., $-(1 \times 2 \times 3)$ and $\frac{1}{1+2+3}$ for the operands $\{1, 2, 3\}$.

Besides, to handle the different forms of numbers, e.g., negative numbers and fractions, MTree also introduces four types of variants $\{n, \frac{1}{n}, -n, -\frac{1}{n}\}$ to denote them, as the example shown in Figure 2.

## 3.2 Problem Encoder

Given a math word problem, we first employ a neural language model to convert the discrete words

---

[1] https://www.sympy.org/
[2] For the operations beyond these four arithmetical operators, such as $a^b$, we follow the processing in (Wang et al., 2022) to convert it to multiple times product.

---

to compact problem representation, and decode the problem representation to MTree, as shown on the left of Figure 2. Two kinds of language models are commonly used in previous works: the recurrent neural networks (RNN-based), e.g., LSTM or GRU, and the pre-trained language models (PLM-based), e.g., BERT or RoBERTa. Inspired by the superior representation ability of PLM, recent works tend to employ PLMs as problem encoder. We follow (Jie et al., 2022; Wang et al., 2022) implementing RoBERTa-base (Liu et al., 2019) or BERT (Devlin et al., 2019) to extract the problem representation. Specifically, given the problem $S = \{w_1, w_2, ..., w_n\}$, containing numerical values $V = \{v_1, v_2, ..., v_m\}$, we concatenate a [CLS] and [SEP] at the beginning and end of the problem text, respectively. Then the output of [CLS] is employed as the entire problem representation. The problem encoding process could be formulated as:

$$E_s, E_V = PLM([CLS]; S; [SEP]), \quad (1)$$

where $E_s$ is the output of [CLS] for the entire problem representation, and $E_V$ denotes the contextual representations of numbers. We also finetune the PLMs during training to make the learned representations better fit the MWP task.

## 3.3 Goal-Driven MTree Generator

Expression tree decoding has been well explored in MWP solving, e.g., sequential decoding with postorder traversal and goal-driven decomposing. Motivated by the intuition and success of goal-driven decomposing (Xie and Sun, 2019), we implement the MTree generator following the top-down decomposition with a goal-driven mechanism. Specifically, we utilize the problem representation ($E_s$ in Equation 1) as the root goal of the MTree, then recursively generate the sequence of sub-goals with the top-down fashion, until the sub-goal is not an operand. The sub-goals are categorized as an explicit token, e.g., operand or operator, by measuring the similarity between sub-goals and candidates. The candidate representations are defined as:

$$E_c = \begin{cases} E_V, & \text{if it is a number} \\ E_{op}, & \text{if it is an operator} \\ E_{con}, & \text{if it is a constant} \\ E_N, & \text{if it is the special token } N_b \end{cases} \quad (2)$$

where $E_V$ is the encoder output in Equation 1, and other $E_*$ are learned embedding. $N_b$ is a special token indicating the end and number of branches of a

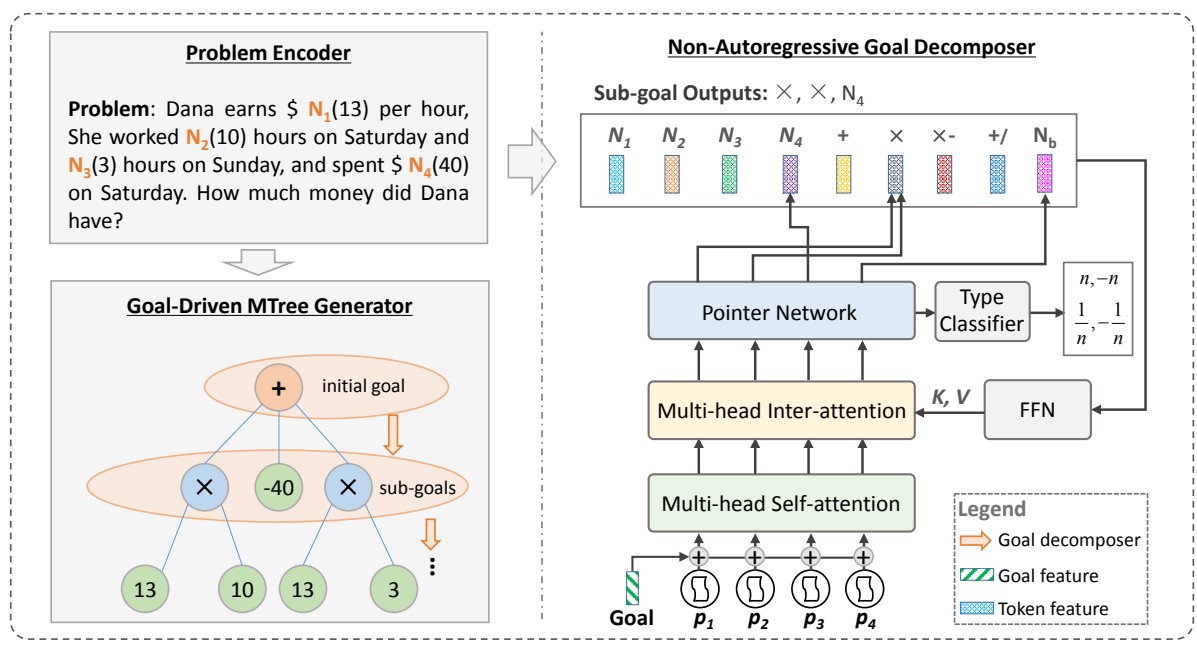

Figure 2: The overall pipeline of our MWP-NAS, which consists of a **problem encoder** and a **goal-driven MTree generator**. To implement top-down decomposing, we devise a **non-autoregressive goal decomposer** (shown on the right side) to implement the multi-branch decomposition.

goal, which will be detailed in next section. Unlike the binary tree where the left and right branches are ordered, the multiple branches (may more than two branches) of our MTree are unordered. Different from (Xie and Sun, 2019) generating left and right sub-goals with different modules, we need to treat all the sub-goals the same and generate them with an identical module.

### 3.4 Non-Autoregressive Goal Decomposer

Towards this end, we design a novel non-autoregressive goal decomposer (NAGD) to handle the unordered multi-branch decomposing as sequence generation, based on the non-autoregressive Transformer (Gu et al., 2018). The pipeline of our proposed NAGD is illustrated on the right in Figure 2. Given the goal representation $E_g$, the NAGD first integrates it with every positional embedding via element-wise sum, and the fused representation is denoted as $E_p$. Following (Vaswani et al., 2017), we embed positions with sine and cosine functions:

$$p_{i,2j} = sin(i/10000^{2j/d_k}), \quad (3)$$

$$p_{i,2j+1} = cos(i/10000^{2j/d_k}), \quad (4)$$

where $i$ denotes the position and $j$ is the $j$-th dimension in $p_i$. To explore relative information and dependencies between positions, we implement a multi-head self-attention as:

$$\tilde{E}_p = \text{MHAtt}(E_p\tilde{W}_Q, E_p\tilde{W}_K, E_p\tilde{W}_V), \quad (5)$$

where $\tilde{W}_Q$, $\tilde{W}_K$, and $\tilde{W}_V$ are learned mapping matrices. To make the communications between sub-goals available, we propose **cross-goal attention** to implement the self-attention of positions. As the example shown in Figure 3(b), when we decompose the left "×" node, we also peek at the information of nodes "−40" and another "×", while the vanilla attention (shown in Figure 3(a)) only capture the information in the left "×" node. For the leaf node −40, we attach dummy child nodes to pass the information to other child nodes. Note that the dummy nodes are only used for cross-goal interaction, and would not be used for loss computation. Through such cross-goal attention, the sub-goals could interact with each other and avoid duplicate decomposition.

We then connect the positions and token candidates by a multi-head inter-attention block, which employs $\tilde{E}_p$ as $Q$, and learns $K$ and $V$ from candidates, as:

$$\hat{E}_p = softmax(\frac{\tilde{E}_p(E_c\hat{W}_K)^T}{\sqrt{d_k}})(E_c\hat{W}_V), \quad (6)$$

where $E_c$ is the representation of target candidates defined in Equation 2. Finally, the contextual representation $\hat{E}_p$ could be used to select the most relevant candidates, *e.g.*, operands or operators, via a pointer network (Vinyals et al., 2015). The probability of position $i$ to choose the $j$-th candidate is

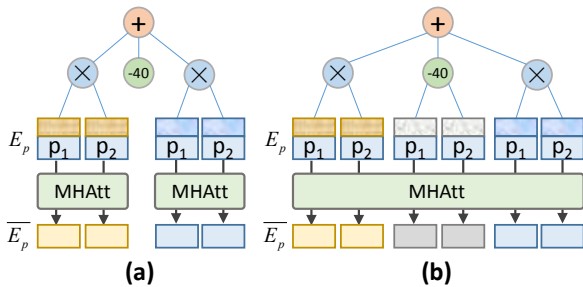

Figure 3: Illustration of vanilla multi-head self-attention (a), and our proposed cross-goal attention (b). MHAtt means multi-head self-attention.

formulated as:

$$\omega_{ij} = u^T \tanh(W_p e_i^p + W_b e_j^c), \quad (7)$$

$$Ptr_i = softmax(\omega_i), \quad (8)$$

where $W_*$ are learned parameters, and $u$ is a column weight vector. $e_i^p$ and $e_j^c$ are the representations of $i$-th position and candidate $j$. $Ptr_i$ is the probabilistic distribution across all the candidates for $i$-th position.

To calculate the loss between predictions and ground-truths, we need to align the positions and the ground-truth sub-goals. We therefore define a pseudo order for ground-truth tokens of each goal. The operators are ahead, followed by the constants and operands in the problem. For the example in Figure 2, the pseudo order of ground-truth sub-goals for node $+$ is $[\times, \times, -40]$. Besides, vanilla non-autoregressive Transformer (Gu et al., 2018) implements fertility prediction to decide the decoding length. In our NAGD, we set a max length as 8 for all the samples, because more than 99% samples in the datasets are with MTree branches less than 8. Meanwhile, we also append a special token $N_b$ at the end of sub-goals (similar with the <END> token in machine translation) to indicate the number of branches for the current goal decomposing, and the loss and prediction after the $N_b$ would be ignored during training and inference, respectively.

After predicting the token of each sub-goal, we need to indicate the type ($\{n, \frac{1}{n}, -n, -\frac{1}{n}\}$ mentioned in Section 3.1) for predicted operands. We design a simple MLP for type classification, and employ focal loss to mitigate the issue of imbalanced classes. The loss is finally integrated with pointer loss to jointly train the whole MWP-NAS.

## 4 Evaluation

Most existing works employ **expression accuracy** and **value accuracy** to evaluate and compare the performance of MWP methods (Xie and Sun, 2019; Wang et al., 2017; Jie et al., 2022). Value accuracy measures the accuracy of the final answer, but fails to evaluate the validity of the solving procedure. Expression accuracy tries to address this issue and measure the exact matching accuracy between predicted and ground-truth expression, and results in much lower performance due to the diverse variants of the same solution expression. Obviously, the expression accuracy should be the same as value accuracy in theory, or slightly lower for some exception cases. The ideal metric to evaluate the expression should be capable of handling the reasonable variants of solution expressions. In other words, all the variants derived from the gold expression with arithmetic rules would lead to the gold answer, and should be considered as true.

However, it is hard to include all the variants and conduct an exhaustive evaluation. Based on the uniqueness of MTree in expression variants, we propose **MTree Accuracy** and **MTree IoU** to evaluate the expression accuracy more accurately. Specifically, following existing expression accuracy to verify the exact matching accuracy on the whole tree, MTree accuracy measures the matching accuracy between predicted and gold unified MTrees. While such evaluation on the whole expression tree fails to measure the partial correctness of expressions, which is also an important way to evaluate the ability of solvers, as well as humans. For example, given the example in Figure 1, $13 \times (10 + 3) + 40$ and $(13 \times 10 + 3 - 40)$ are two false expressions, and the former one only uses the wrong operation "+" for the expense, but all the items are correct. While the latter applies incorrect calculation of income on Sunday, which should obtain a lower score. To this end, we propose MTree IoU to calculate the accuracy of paths connecting root and leaves to measure the partial correctness of expressions. MTree IoU first transforms the predicted and ground-truth expressions to MTrees, then constructs root-leaf path sets $P_p$ and $P_g$ for them, respectively. Inspired by the evaluation in object detection (Ren et al., 2015), we calculate the Intersection over Union (IoU) as:

$$MTreeIoU = \frac{|P_p \cap P_g|}{|P_p \cup P_g|}, \quad (9)$$

where $|\cdot|$ denotes the size of the set. The overall MTree IoU of test set is averaged over all the test samples. Note that there may exist duplicate paths in an MTree due to the duplicate numbers and multiple usages, such as the number "13" in Figure 1 is used multiple times in the MTree. We treat such duplicate paths as different paths in the IoU calculation, because if the solver only predicts one of them, it should not be totally correct. In our codes, we actually use 'List' as the type of $P_p$ and $P_g$ in Equation 9, rather than 'Set' that may ignore the duplicated paths.

## 5 Experiments

### 5.1 Datasets

To evaluate the effectiveness of the proposed MWP-NAS and metrics, we conduct extensive experiments on Math23K and MAWPS, and compare with SoTAs. Math23K (Wang et al., 2017) contains 23162 arithmetical problems. For fair comparison, we use the splits following (Wang et al., 2022; Zhang et al., 2020), resulting in 21162, 1000, and 1000 for training, validation, and test, respectively. MAWPS (Koncel-Kedziorski et al., 2016) is a much smaller MWP dataset, which contains 2373 samples. We follow (Wang et al., 2022) to preprocess for MTree and obtain 2163 samples. Due to its small size, following previous works (Zhang et al., 2020; Jie et al., 2022), we conduct 5-fold cross-validation on MAWPS, resulting in 433 samples for four folds and 431 samples for another fold. We report the averaged results across five folds.[3]

### 5.2 Experimental Settings

We implement the proposed MWP-NAS with PyTorch and conduct experiments with an NVIDIA RTX A6000 GPU. The maximum size of MTree branch is set as 8, because more than 99% samples in the datasets have fewer than 8 child nodes. We fine-tune the pre-trained language models, *i.e.*, RoBERTa/BERT-base, during training and initialize the global learning rate as $2e^{-5}$ and $5e^{-5}$ for Math23K and MAWPS, respectively. The hidden

---

[3]In the main part of (Jie et al., 2022), the authors implement a different training setting, **combining training and validation sets for training**, and run five times with different random seeds for averaging on Math23K. The 5-fold split of MAWPS also has two different settings: the split in (Wang et al., 2022) and others. For fair and comprehensive comparisons, we implement our MWP-NAS on these two splittings, and term them as *Reasoner Split* and *SUMC Split* in Table 1. The main experiments (including the ablations) are based on SUMC split.

Table 1: Performance comparison with baselines. $\diamondsuit$ means the results referred from (Wang et al., 2022). $\spadesuit$ means the results based on *SUMC split*. $\clubsuit$ indicates the results of the data split in (Jie et al., 2022), which implements a 5-fold CV train-test setting for Math23K (we call it *Reasoner Split*).

| Model | Math23K | MAWPS |
|---|---|---|
| Seq2Seq (Wang et al., 2017) | 58.1 | 59.5 |
| T-RNN (Wang et al., 2019) | 66.9 | 66.8 |
| GROUP-ATT (Li et al., 2019a) | 69.5 | 76.1 |
| GTS (Xie and Sun, 2019) | 75.6 | 82.6 |
| Graph2Tree (Zhang et al., 2020) | 77.4 | 83.7 |
| NeuralSymbolic (Qin et al., 2021) | 75.7 | - |
| HMS (Lin et al., 2021) | 76.1 | 80.3 |
| NUMS2T (Wu et al., 2021) | 78.1 | - |
| BERT-Tree (Li et al., 2022) | 82.4 | - |
| SAU-Solver (Qin et al., 2020) | 76.2$^{\diamondsuit}$ | 75.5$^{\diamondsuit}$ |
| UniLM (Dong et al., 2019) | 77.5$^{\diamondsuit}$ | 78.0$^{\diamondsuit}$ |
| DeductReasoner (Jie et al., 2022) | 84.3$^{\spadesuit}$ | 86.0$^{\spadesuit}$ |
| DeductReasoner (Jie et al., 2022) | 85.1$^{\clubsuit}$ | 91.2$^{\clubsuit}$ |
| SUMC-Solver (Wang et al., 2022) | 82.5 | 82.0 |
| MWP-NAS (SUMC Split) | **84.8**$^{\spadesuit}$ | **86.7**$^{\spadesuit}$ |
| MWP-NAS (Reasoner Split) | **86.1**$^{\clubsuit}$ | **91.4**$^{\clubsuit}$ |

size of our non-autoregressive goal decomposer is 768 for every layer. We run 1000 and 2000 epochs on Math23K and MAWPS for training, and choose the checkpoints with the best performance on validation set for test.

### 5.3 Comparisons with Baselines

We first evaluate and analyze the effectiveness of our proposed non-autoregressive solver for MTree by comparing its performance with the state-of-the-arts. Here we only compare the accuracy of the final answer and leave the expression accuracy evaluation in the next section, because of two points: (1) The expression generated by our MWP-NAS based on MTree could be written as multiple reasonable expression variants, which is hard to evaluate the expression accuracy, and (2) only a few numbers of previous works reported the expression accuracy, it would be many N/A in the comparison. As the results shown in Table 1, our proposed MWP-NAS outperforms all the baselines and sets a new state-of-the-art on both datasets, which demonstrates the effectiveness and superiority of our method. As the only two methods based on MTree, the SUMC-Solver employs codes prediction of leaf node to reconstruct the expression tree, which performs much lower than our MWP-NAS. The reason might be that the code prediction of leaf nodes makes the nodes independent and break the arithmetical relation between the nodes. While our MWP-NAS implementing attention mechanism is able to ex-

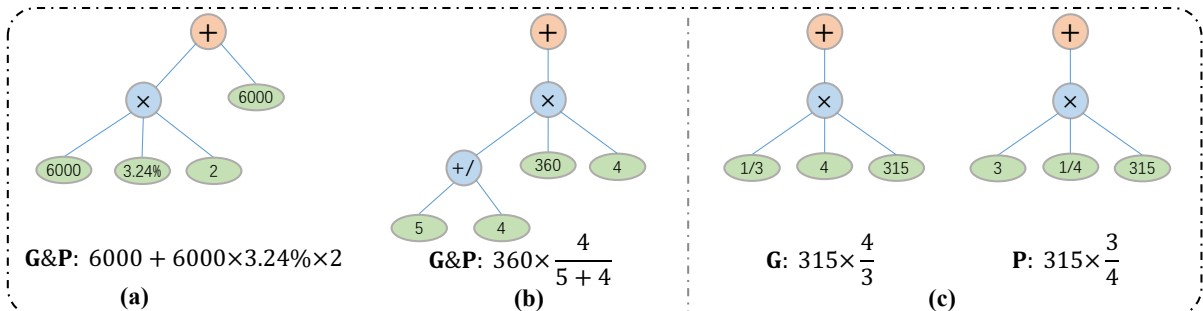

Figure 4: Several cases of our MWP-NAS with MTree structure. **G** and **P** indicate the **Ground truth** and **Prediction**, respectively. We merge the ground truth and prediction for correct predictions, *i.e.*, (a) and (b), and illustrate a failure case in (c).

Table 2: The results of ablation study of cross-goal attention on Math23K and MAWPS.

| Model | Math23K | MAWPS |
|---|---|---|
| w/o Cross-Goal Attention | 84.1 | 86.1 |
| with Cross-Goal Attention | 84.8 | 86.7 |

plore and capture the relations between numbers, and yield better results. Besides, we also note De-ductReasoner (Jie et al., 2022), implementing complex relation modeling and deductive reasoning, performs very close to our work for both data split settings, which may imply that introducing the deductive reasoning to MTree structure would bring some new insights.

### 5.4 Effectiveness of Cross-Goal Attention

We also conduct ablation studies to investigate the effectiveness of the proposed cross-goal attention, and show the results in Table 2. The results demonstrate that equipped with our cross-goal attention, our model gains significant improvement, *e.g.*, from 84.1 to 84.8 on Math23K. This suggests that with such a cross-goal attention mechanism, the information belonging to different goals could be passed and aggregated. Such cross-goal information integration apparently improves the accuracy of single-goal decomposition and benefits the expressiveness of the model overall.

### 5.5 Analysis on Different Branch Numbers

In MWP solving, complex problems always contain more operands and result in more branches in MTree. To investigate the performance on different difficult problems, we test and compare the value accuracy with different branch numbers. We first define the **branch number** of an MTree as the max-

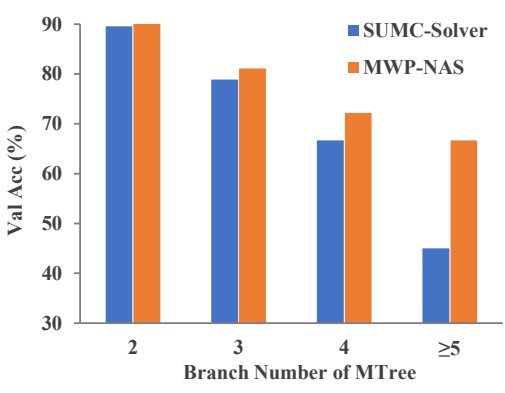

Figure 5: Performance comparison on different branch numbers of MTree.

imum number of branches in it. For example, the branch number of the MTree in Figure 1 is three, because the "+" node is the one with the most branches in it and has three branches. The experimental results are illustrated in Figure 5. From the results, we can observe that our MWP-NAS outperforms the SUMC-Solver at all the branch numbers, and exhibits remarkable boosting at larger branch numbers, which means that our MWP-NAS works better than SUMC-Solver for more complicated problems. Besides, with the branch number increasing, both methods show a decreasing trend for value accuracy, which exhibits more inferior performance for more difficult problems and is consistent with our common sense.

### 5.6 Illustration with MTree Structure

To comprehensively probe the proposed MWP-NAS, we also visualize several solving cases with MTree structure, as illustrated in Figure 4. We observe that our MWP-NAS is capable of generating accurate MTree structure, even for the complex

Table 3: Evaluation comparison with our proposed **MTree Acc** and **MTree IoU**. We reproduce and evaluate all the results based on the released codes.

| Model | Exp Acc | Val Acc | MTree Acc | MTree IoU |
|---|---|---|---|---|
| Seq2Seq (Wang et al., 2017) | 51.7 | 58.3 | 58.2 | 62.99 |
| GTS (Xie and Sun, 2019) | 64.4 | 75.7 | 75.3 | 82.77 |
| Graph2Tree (Zhang et al., 2020) | 66.0 | 77.4 | 76.9 | 83.53 |
| DeductReasoner (Jie et al., 2022) | 76.4 | 84.3 | 83.6 | 88.86 |
| SUMC-Solver (Wang et al., 2022) | - | 82.9 | 82.4 | 87.26 |
| MWP-NAS (Ours) | - | 84.8 | 83.8 | 89.73 |

**Problem:** The fruit shop sells bananas, oranges and pineapples for a total of 150 kg on Sunday, of which bananas are 27.5 kg, and the number of oranges sold is 3.6 times that of bananas. How many kilograms of pineapples are sold in the fruit shop on Sunday?
**Ground-Truth Expression**: $150-27.5\times3-27.5$   **DeductReasoner**: $150-27.5-27.5\times3$   **Exp Acc**: ✘ (False)   **MTree Acc**: ✔ (True)

**Problem:** There are 36 chickens, and the number of ducks is twice that of chickens. How many chickens and ducks are there?
**Ground-Truth Expression**: $(1+2)\times36$   **DeductReasoner**: $36\times2+36$   **Exp Acc**: ✘ (False)   **MTree Acc**: ✔ (True)

Figure 6: Examples evaluated by Exp Acc and our proposed MTree Acc, predicted with DeductReasoner. Traditional Exp Acc fails to capture the same mathematical semantics with different expressions, while our MTree Acc successfully handles this.

arithmetic expressions. For example, as shown in Figure 4(b), it could easily capture the composition between 5 and 4. From the exemplar shown in Figure 4(c), we note our MWP-NAS may fail to predict right type for operands, which may because the optimization objectives of number type is an extra loss. This also motivates us keep exploration of this direction for further improvements.

### 5.7 Evaluation on MTree Metrics

To study the effectiveness of our proposed MTree-based metrics, we implement 5 representative MWP methods with the open source codes and compare them with our MWP-NAS with more metrics on Math23K. The comparison results are illustrated in Table 3. As aforementioned, the expression accuracy should be consistent with the value accuracy, while it is much lower than value accuracy in practice. The MTree accuracy measures the performance by transforming the ground-truth expression and prediction to the unified MTree structure, demonstrating slightly lower than the value accuracy. We visualize several examples in Figure 6 to check whether the predicted expression is right and indicated as wrong by expression accuracy. We observe that MTree accuracy is able to handle different variants derived from ground-truth, *e.g.*, commutative law for the first case and distributive law for the second one in Figure 6. We have also checked the samples (only six samples in test set) failed by MTree accuracy but the final value is right, and found that three of them are annotated

wrong ground-truth expressions, and two of them give the wrong expression but result in the gold answers. For the MTree IoU, it further evaluates the expressions with local paths, and is able to assess the partial correctness of an expression.

## 6 Conclusion

We presented a novel non-autoregressive MWP solver (MWP-NAS) based on the unified MTree structure. Our proposed MWP-NAS implements a goal-driven manner for multi-branch decomposition to generate the MTree of expressions. We devised a cross-goal attention strategy to pass information between goals during goal decomposing. We have also designed two metrics based on MTree for better expression evaluation. Experiments conducted on Math23K and MAWPS demonstrated the effectiveness of our approach and metrics.

## 7 Acknowledgement

This research is supported by A*STAR, CISCO Systems (USA) Pte. Ltd and National University of Singapore under its Cisco-NUS Accelerated Digital Economy Corporate Laboratory (Award I21001E0002). This research is also partially supported by the National Natural Science Foundation of China under grant 62102070, 62220106008, and U20B2063. This research is also partially supported by Sichuan Science and Technology Program under grant 2023NSFSC1392.

## 8 Limitations

To obtain the MTree, we need first to simplify and expand the original expression using `sympy.simplify` and `sympy.expand` of SymPy package. Some cases (about 0.2 percent, *i.e.*, 2 cases in test set) are failed to automatically transform to MTree, due to the expression unification. We discard them during training and use the original form as ground-truth for test evaluation. This makes our reported performance decrease by 0.2 percent. Therefore, automatically unifying the expressions could be a small issue and limitation of our work. Besides, with the blooming of large language models (LLMs), we only conducted experiments with some relatively small models, *i.e.*, BERT and RoBERTa, for fair comparison, and ignored the integration of MTree and LLMs at the moment, which also could be a limitation of our work and will be explored in the future.

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

# A Appendix

## A.1 Refine the MTree Structure

In this part, we attempt to provide an under-mature refinement for MTree structure and hope to encourage more research attention and insights on unifying the expression representation. As mentioned in

Table 4: Results and comparisons with preliminary refined MTree (RefMTree).

| Model | Val Acc | MTree Acc | MTree IoU |
|---|---|---|---|
| SUMC-Solver | 82.9 | 82.4 | 87.26 |
| -RefMTree | 79.8 | 79.5 | 86.99 |
| MWP-NAS | 84.8 | 83.8 | 89.73 |
| -RefMTree | 84.3 | 83.4 | 89.71 |

Section 3.1, in the design of original MTree (Wang et al., 2022), there needs to introduce additional indicator to denote the form of numerical values, $\{n, \frac{1}{n}, -n, -\frac{1}{n}\}$ in specific. We observe that it is possible to omit the form indicator with the new operators $\{\times-, +/\}$. Rethinking the operation for $\times-$, it calculates the opposite value of the product of the operands. Suppose we have only one operand $n$ for $\times-$, the result could be $-n$. Similar transformations also can be implemented to obtain $\frac{1}{n}$ with $+/$, and $-\frac{1}{n}$ with the sequential combination of $+/$ and $\times-$. With such modification, the MTree in Figure 2 only needs to revise "-40" to "40" and add an internal node $\times-$ between 40 and the root. We also conduct several preliminary experiments with the refined MTree (we call it RefMTree) on Math23K. The experimental settings are the same as the ones in Section 5.2, except for removing the type classifier of MWP-NAS and the form code of SUMC-Solver. From the results shown in Table 4, we observe that all the performances are decreased, which suggests that such simple and naive modification for MTree refinement is far from good enough. We also note that our MWP-NAS decreases slightly, while the SUMC-Solver gets significant performance drop. We assume it may be that MTree refinement changed the format of path codes of SUMC, and led to inferior results. Though the additional internal nodes deepen the MTree and make our MWP-NAS to predict the MTree more difficult, MWP-NAS still benefits from removing the type classifier and yields slightly lower results. In summary, the simple refinement of MTree does not work well for both MTree-based methods, and encourages us to keep the exploration in future.

## A.2 Complementary Experimental Results

MWP solving is a fundamental task for evaluating the reasoning ability of language models, therefore there exist many benchmark datasets, such as Math23K (Wang et al., 2017), MAWPS (Koncel-Kedziorski et al., 2016), MathQA (Amini et al., 2019), MATH (Hendrycks et al., 2021), AS-Div (Miao et al., 2020), and SVAMP (Patel et al., 2021). To comprehensively evaluate the effectiveness of our proposed MWP-NAS, we also conduct complementary experiments on more datasets, MathQA and SVAMP [4] in specific[5]. The results are reported in Table 5, from which we observe that the comparison results across different methods show inconsistency on two datasets, even exhibiting significant contradictions for some comparisons. For example, GROUP-ATT (Li et al., 2019a) outperforms Graph2Tree (Zhang et al., 2020) on MathQA by 0.9, but shows remarkable performance decrease, *i.e.*, 15.0, on SVAMP. We also observe that our MWP-NAS outperforms almost all the baselines on these datasets, except for the Graph2Tree (Zhang et al., 2020) on SVAMP. This may come from the graph representations, *e.g.*, Quantity Cell Graph and Quantity Comparison Graph, which are good at handling the problem variants in SVAMP dataset. When we go beyond this aspect, comparing with the strong baseline DeductReasoner (Jie et al., 2022), our MWP-NAS substantially outperforms it and achieves the state-of-the-art performance, which further verifies the effectiveness of out proposed MWP-NAS.

Table 5: Performance comparison with baselines on two extra datasets, MathQA and SVAMP. ♠ means the results referred from (Li et al., 2022), and ♣ indicates the results referred from (Jie et al., 2022), otherwise from the original paper.

| Model | MathQA | SVAMP |
|---|---|---|
| Seq2Seq (Wang et al., 2017) | - | - |
| T-RNN (Wang et al., 2019) | - | - |
| GROUP-ATT (Li et al., 2019a) | 70.4♠ | 21.5♣ |
| GTS (Xie and Sun, 2019) | - | 30.8♣ |
| Graph2Tree (Zhang et al., 2020) | 69.5♣ | **36.5♣** |
| NeuralSymbolic (Qin et al., 2021) | - | - |
| HMS (Lin et al., 2021) | - | - |
| NUMS2T (Wu et al., 2021) | - | - |
| BERT-Tree (Li et al., 2022) | 73.8♣ | 32.4♣ |
| SAU-Solver (Qin et al., 2020) | - | - |
| UniLM (Dong et al., 2019) | - | - |
| DeductReasoner (Jie et al., 2022) | 80.6 | 35.3 |
| SUMC-Solver (Wang et al., 2022) | - | - |
| MWP-NAS | **81.2** | 35.5 |

---

[4] For MAWPS and SVAMP, we use BERT-base as the problem encoder.

[5] We do not include the results on these two datasets in the main part because only a few baselines conducted experiments on them. Similarly, we did not conduct experiments on the other datasets due to the less baselines on them. Therefore we only regard these results as a complementary to the main results reported in Table 1.