# OpenReview forum: "Non-Autoregressive Math Word Problem Solver with Unified Tree Structure"
_EMNLP/2023/Conference — EMNLP 2023 Main_

### Official Review · Reviewer_P6mx · 2023-08-03

**Soundness:** 4

**Excitement:**

4: Strong: This paper deepens the understanding of some phenomenon or lowers the barriers to an existing research direction.

**Paper Topic And Main Contributions:**

This paper proposes a method for math word problem-solving. It is a goal-driving method that starts from the root of the expression tree and decodes the tree layer by layer to the leaf nodes. It extends this approach by 1. predicting all child nodes in the next layer in parallel and 2. using cross-attention to make the parallel prediction aware of each other.

**Reasons To Accept:**

1. The method proposed by this paper is interesting and sound. It combines goal-driving, MTree (a unified, unique representation of math equations), and cross-attention methods in harmony.
2. The proposed evaluation method is a more fine-grained metric than the expression and result accuracy.

**Reasons To Reject:**

A. It is not clear how to handle the order of the prediction in each sub-goal, which seems to be crucial for the result. More specifically, in the left subfigure of Figure 2, why is p1 connected to "x" and p3 connected to "N4", can we change their order?


**Reproducibility:**

3: Could reproduce the results with some difficulty. The settings of parameters are underspecified or subjectively determined; the training/evaluation data are not widely available.

**Reviewer Confidence:**

4: Quite sure. I tried to check the important points carefully. It's unlikely, though conceivable, that I missed something that should affect my ratings.

**Typos Grammar Style And Presentation Improvements:**

The writing of Section 3.3 and Section 3.4 could be improved. You may first introduce what p1 to p4 means and how they are connected to outputs before introducing the details. In lines around 339 where you introduce the positional embeddings, the readers don't know what are there in each position.
The sentence between lines 400 to 402 is hard to understand.
$E_s$ in Equation (1) is introduced as $E_p$ in line 299.

---

> ### Author Rebuttal · Authors · 2023-08-28
>
> Thanks for the endeavors and efforts put into reviewing our paper. We're grateful to all the valuable reviews. In this rebuttal, we detailly answer the reviewer’s questions one-by-one, and hope the responses clear up all the concerns.
>
>
> **Q1:** It is not clear how to handle the order of the prediction in each sub-goal, which seems to be crucial for the result. More specifically, in the left subfigure of Figure 2, why is p1 connected to "x" and p3 connected to "N4", can we change their order?
> >**Response:** Thanks for the review. As described in L386-393, we define a **pseudo-order for the subgoals** to align positions and subgoals during training. Specifically, in this work we make the operators are ahead and followed by the constants and operands. Following such pseudo-order rule, $p_1$ connects to ‘$\times$’ and $p_3$ connects to ‘$N_4$’. With certain rule, the order cannot be changed, but we can also define other rules for the pseudo-order.
>
> **Q2:** The writing of Section 3.3 and Section 3.4 could be improved. You may first introduce what p1 to p4 means and how they are connected to outputs before introducing the details. In lines around 339 where you introduce the positional embeddings, the readers don't know what are there in each position. The sentence between lines 400 to 402 is hard to understand. E_s in Equation (1) is introduced as E_p in line 299.
> >**Response:** We apologize for the unclear description. We will improve these parts as the reviewer suggested, and conduct thorough proof-reading over the whole paper. For $E_s$ and $E_p$ in Eq(1) and L299, it should be $E_s$ in L299, we will revise this in the revision. Thanks for the suggestive and valuable comments.

---

### Official Review · Reviewer_jd6M · 2023-08-03

**Soundness:** 4

**Excitement:**

4: Strong: This paper deepens the understanding of some phenomenon or lowers the barriers to an existing research direction.

**Missing References:**

Measuring Mathematical Problem Solving With the MATH Dataset, Hendrycks et al., NeurIPS 2021

A Diverse Corpus for Evaluating and Developing English Math Word Problem Solvers, Miao et al. ACL-2020

Are NLP Models really able to Solve Simple Math Word Problems? Patel et al., ACL-2021

**Paper Topic And Main Contributions:**

This paper presents a non-autoregressive approach to deducing the multiple solution variants (i.e., valid solutions for the same MWP but formulated as different expressions) based on a unified tree (in which the unified tree presents a solution expression where nodes in a tree are permutable are identical for all expression variants). The authors also propose a cross-goal attention mechanism to leverage the information across goals to enhance the model's robustness. Furthermore, the authors introduce a partial-tree evaluation metric to assess the capability of an MWP solver. Experimental results demonstrate the effectiveness of the proposed approaches.

**Questions For The Authors:**

Q1: Lines #322-#323: What is the effect of the special token Nb? Is Nb a predefined constant (Lines #399-#401)? Do all MWPs share the same Nb? How to decide the branch number in the inference? Related details are unclear.

Q3: Line #353: Does cross-goal attention only consider other subgoals under the same parent goal? Or will it also consider the parent subgoal, grandparent subgoal, even the main goal?

Q3: Lines #453-#460: Although the MTree IoU assesses the partial correctness of the expression, are there other contributions to MWP solving? For example, can it be used to improve the model performance?

Q4: Line #465-#467: By Eq(9), the proposed IoU formula can't directly distinguish two different paths with duplicated numbers. Thus, the assessment may be overestimated. How to handle such cases?

**Reasons To Accept:**

Unlike previous work on goal-driven tree decoders, the proposed method is like the human's problem-solving strategy, that is, to generate subgoals by the expansion from top (main-goal) to bottom (subgoals) (rather than from left to right). Experimental results support the claim of the proposed cross-goal attention mechanism to enhance the system performance. This work and experimental results will benefit future research on MWP solving.

**Reasons To Reject:**

(1) One of the motivations for introducing MTree is to address the challenge of learning efficiency issues for handling multiple solution variants. However, the authors didn’t provide related experimental results to support this claim.

(2) Lack of assessments for publicly challenging MWP datasets (such as Math, SVAMP, or ASDIV).

**Reproducibility:**

3: Could reproduce the results with some difficulty. The settings of parameters are underspecified or subjectively determined; the training/evaluation data are not widely available.

**Reviewer Confidence:**

4: Quite sure. I tried to check the important points carefully. It's unlikely, though conceivable, that I missed something that should affect my ratings.

**Typos Grammar Style And Presentation Improvements:**

(1)	Lines #031: What is “xxx”? It is suggested to revise the statement if “xxx” represents “to be released upon acceptance.”

(2): Line #299: Is E_p or E_s(in Eq(1)) ?

(3) Table 1: The same notations but are different data splitting methods for ReduceReasoner models. It is suggested to use different notations to distinguish them.

---

> ### Author Rebuttal · Authors · 2023-08-28
>
> Thanks for the endeavors and efforts put into reviewing our paper. We're grateful to all the valuable reviews. In this rebuttal, we detailly answer the reviewer’s questions one-by-one, and hope the responses clear up all the concerns.
>
> **Q1:** Lines #322-#323: What is the effect of the special token Nb? Is Nb a predefined constant (Lines #399-#401)? Do all MWPs share the same Nb? How to decide the branch number in the inference? Related details are unclear.
>
> >**Response:** We apologize for the ambiguous description. The special token $N_b$ is somehow like the **“<END>”** token in traditional text generation task, which is shared by all the MWPs. **$N_b$ is not a predefined constant**, it is a token that is appended at the end of branches for the same goal/subgoal. It does not directly present the number of branches of a goal, but it can indicate the paragraph length indirectly, because all the branches must be setting before it. As mentioned in L396-402, during the inference, we set a max number, 8 in this work, for parallel decoding of branches, and ignore the predictions after the generated $N_b$ token. We will make description more detail and clear in the revision. Thanks for your suggestive reviews.
>
>
> **Q2:** Line #353: Does cross-goal attention only consider other subgoals under the same parent goal? Or will it also consider the parent subgoal, grandparent subgoal, even the main goal?
>
> >**Response:** Thanks for your comments. As shown in Figure 3(b), the cross-goal attention considers **all the subgoals in the same depth** of the MTree, which could belong to different parent goals. The cross-goal attention makes the interactions between goals available and avoids wrong decomposition. For the example in Figure 3(b), there exist two '$\times$' goals in the second layer, which could share very similar latent representations because they belong to the same '$\times$' token class. If we decompose them independently, it may result in the same child nodes and induce repetition. When we employ cross-goal attention, the decomposition process of left '$\times$' would peek at the information of the right '$\times$' and interact with it. Then it could avoid to generate the same child nodes with the right '$\times$', and vice versa for the decomposition of right '$\times$'.  In this paper, we did not consider the parent subgoal, grandparent subgoal, and the main goal here, because the motivation of cross-goal attention is to make the interactions across goals available. **The interactions across parent subgoals could be explored during the grandparent subgoal decomposition**, and so on. Hope our response is clear and helpful.
>
> **Q3:** Lines #453-#460: Although the MTree IoU assesses the partial correctness of the expression, are there other contributions to MWP solving? For example, can it be used to improve the model performance?
> >**Response:** Thanks for your suggestive review. In this work, we only employ MTree IoU as a metric to assess the partial correctness of the expression. We believe it could be used to improve the model performance, such as **adding an optimization goal based on MTree IoU** to improve the MTree IoU during training, it could make the discrete/binary score of traditional accuracy of single expression to a continuous score that is easy to optimize.
>
> **Q4:** Line #465-#467: By Eq(9), the proposed IoU formula can't directly distinguish two different paths with duplicated numbers. Thus, the assessment may be overestimated. How to handle such cases?
> >**Response:** Thanks for your review. Yes, the vanilla IoU formula in Eq(9) can't directly distinguish two different paths with duplicated numbers. To address this issue, **as described in L465-467,** we treat the duplicated paths as different paths. **In our code implementation, we actually use ‘List’ as the type of $P_p$ and $P_g$ in Eq(9), rather than ‘Set’ that may ignore the duplicated paths.** We apologize for the unclear description, and will add the implementation details in the revision.
>
> **Clarifications of Reasons to Reject:**
>
> **Q5:** One of the motivations for introducing MTree is to address the challenge of learning efficiency issues for handling multiple solution variants. However, the authors didn’t provide related experimental results to support this claim.
> >**Response:** Thanks for the review. We assume the reviewer means the claim about **‘effectively’ in L15 in Abstract.** First we apologize for the unclear description. Actually we want to claim that it is hard to learn the ‘mapping function’ between input and output domain, because a problem might has multiple variants for the solution expression. This will push the model to learn multiple different outputs for the same input, which violates the **exactly one output rule** in the definition of function [1]. So it will make the model hard to optimize and might lead to inferior performance, which we claimed ```“making it hard for the model to learn the mapping function between the input and output spaces effectively”``` in Abstract. To verify the effectiveness of mapping function learning, we think the performance could be the straightforward way, and will try other perspecitves to evaluate the effectiveness. Hope this makes our motivation clearer.
> >>[1] The definition of function from Wikipedia: a function from a set X to a set Y assigns to each element of X **exactly one element** of Y. https://en.wikipedia.org/wiki/Function_(mathematics)
>
> **Other issues:**
>
> **Q6:** Lines #031: What is “xxx”? It is suggested to revise the statement if “xxx” represents “to be released upon acceptance.”
> >**Response:** Thanks for the suggestion. We use ‘xxx’ as placeholder for code link that will be available after double-blind review. We apologize for the informal presentation.
>
> **Q7:** Line #299: Is E_p or E_s(in Eq(1)) ?
> >**Response:** We apologize for the carelessness. It should be $E_s$ in L299.
>
> **Q8:** Lack of assessments for publicly challenging MWP datasets (such as Math, SVAMP, or ASDIV)
> >**Response:** Thanks for your constructive review. As many baselines, such as Seq2Seq, GTS, Graph2Tree, and DeductReasoner, conduct experiments on Math23k and MAWPS, we also choose them for fair comparison. For the benchmarks you mentioned, we will try to conduct experiments on them and compare with the baselines in the revision.
>
> For other minor issues that are not listed above, e.g., missing references and notation ambiguous, we will address them in the revision, and will conduct thorough proof-reading to make the paper clearer. Thanks again for the valuable reviews. Hope the above responses are clear and helpful, and could address your concerns.

---

### Official Review · Reviewer_fZF3 · 2023-08-05

**Soundness:** 3

**Excitement:**

3: Ambivalent: It has merits (e.g., it reports state-of-the-art results, the idea is nice), but there are key weaknesses (e.g., it describes incremental work), and it can significantly benefit from another round of revision. However, I won't object to accepting it if my co-reviewers champion it.

**Paper Topic And Main Contributions:**

The paper proposes a unified tree structure for the expression representation where the elements are permutable and identical for all expression variants while decoding solution for a MWP. They propose a technique to parse the MWP and generate an expression for the MWP in unified tree structure.

The main contributions of the paper are as follows:
1. Authors aim to solve and important problem of representing an ambiguous correct expression for a MWP.
2. Authors propose a non-autoregressive tree decoding strategy which does goal driven decoding using the unified tree structure.
3. Authors also propose two new metrics for evaluation the expression tree MtreeAcc and MtreeIoU

**Reasons To Accept:**

1. Ambiguous but correct expression tree for MWP is a very real problem and affect the performance of MWP solvers, so a the idea of using a unified tree structure for expressions is an interesting idea.
2. The proposed technique achieves best performance on Math23K dataset.
3. Overall the paper is well written and clear.

**Reasons To Reject:**

- The performance of the proposed approach on other datasets is missing from the paper.
- Comparison of performance with LLMs is missing from the paper.
- Qualitative analysis of the unified tree structure is important but it is missing from the paper.

**Reproducibility:**

3: Could reproduce the results with some difficulty. The settings of parameters are underspecified or subjectively determined; the training/evaluation data are not widely available.

**Reviewer Confidence:**

4: Quite sure. I tried to check the important points carefully. It's unlikely, though conceivable, that I missed something that should affect my ratings.

---

> ### Author Rebuttal · Authors · 2023-08-29
>
> Thanks for the endeavors and efforts put into reviewing our paper. We're grateful to all the valuable reviews. In this rebuttal, we detailly answer the reviewer’s questions one-by-one, and hope the responses clear up all the concerns.
>
> **Q1:** The performance of the proposed approach on other datasets is missing from the paper.
> >**Response:** Thanks for the suggestive review. As most representative baselines conduct experiments on Math23k and MAWPS, such as seq2seq, Graph2Tree, GTS, DeductReasoner, and SUMC-Solver, we choose these two benchmarks to verify the effectiveness of our approach for fair comparison. As the reviewer mentioned, we also note some methods conduct experiments on other datasets, e.g., MathQA and SVAMP, and will conduct experiments on these datasets for comprehensive comparison.
>
>
>
> **Q2:** Comparison of performance with LLMs is missing from the paper.
> >**Response:** Thanks for the suggestion. LLMs demonstrate superb generation and emergent ability in many NLP tasks, as well as MWP solving. We compare with LLMs referring the results of famous Chain-of Thought paper on MAWPS dataset. The results are as below:
> \begin{matrix}
> \hline
> \text{UL2 20B} & \text{LaMDA 137B} & \text{GPT-3 175B} & \text{Codex} & \text{PaLM 540B} & \text{ChatGPT} & \text{MWP-NAS (121M)} \\\\ \hline
> 19.1 & 57.9 & 87.1 & 92.6 & 93.3 & 95.7 & 86.7 \\\\ \hline
> \end{matrix}
>
> >We note that with huge number of parameters (various from tens billion to hundreds billion), the LLMs demonstrate different performance. GPT-3, Codex, PaLM, and ChatGPT with strong emergent ability could solve the MWPs with high accuracy, while smaller models such as UL2 and LaMDA cannot perform well in the Chain-of-Thought scenario (with few-shot learning). Our MWP-NAS only has 121M parameters (including 110M of BERT-base) could achieve considerable performance. We must admit that the LLMs don’t need additional training for different datasets, while our small MWP-NAS needs such training. To make our MTree structure benefit from the LLMs, we will try to integrate the MTree structure with LLMs prompt learning to explore more potentials in future work. Thanks again for your constructive and valuable reviews.
>
> **Q3:** Qualitative analysis of the unified tree structure is important but it is missing from the paper.
> >**Response:** Thanks for the suggestion. We present some qualitative analyses for better evaluation based on MTree in Figure 5, in Page 8, which verifies the superiority of evaluation based on unified tree structure (here we have not directly exhibited the case with tree structure). We will visualize some cases with tree structure of predicted MTree in the revision to qualitatively analyze the unified tree structure, and explore more potential points with failure case study.

---

### Meta-Review · Area_Chair_rQrQ · 2023-09-18

**Recommendation:** 4

**Metareview:**

Paper Topic And Main Contributions:
* This paper proposes a method for math word problem-solving. It is a goal-driving method that starts from the root of the expression tree and decodes the tree layer by layer to the leaf nodes.
* It extends this approach by 1. predicting all child nodes in the next layer in parallel and 2. using cross-attention to make the parallel prediction aware of each other.
* The authors introduce a partial-tree evaluation metric to assess the capability of an MWP solver.
* Experimental results demonstrate the effectiveness of the proposed approaches.

Reasons to accept:
* The proposed method is interesting and sound. It combines goal-driving, MTree (a unified, unique representation of math equations), and cross-attention methods.
* The proposed evaluation methods are valuable; they are more fine-grained than expression and result accuracy.

Reasons to reject:
* The performance of the proposed approach on other datasets (such as Math, SVAMP, or ASDIV) is missing. The authords explain that the datasets they do use (Math23k and MAWPS) are used by many of the baselines. Experiments on MathQA and SVAMP will be added to the final version.
* Comparison of performance with LLMs is missing. The authors address this point in the rebuttal.
* Qualitative analysis of the unified tree structure is missing. The authors point to existing qualitiative analysis and will add further visualization.

---

### Decision · Program_Chairs · 2023-10-07

**Decision:**

Accept-Main

**Comment:**

Paper Topic And Main Contributions:
* This paper proposes a method for math word problem-solving. It is a goal-driving method that starts from the root of the expression tree and decodes the tree layer by layer to the leaf nodes.
* It extends this approach by 1. predicting all child nodes in the next layer in parallel and 2. using cross-attention to make the parallel prediction aware of each other.
* The authors introduce a partial-tree evaluation metric to assess the capability of an MWP solver.
* Experimental results demonstrate the effectiveness of the proposed approaches.

Reasons to accept:
* The proposed method is interesting and sound. It combines goal-driving, MTree (a unified, unique representation of math equations), and cross-attention methods.
* The proposed evaluation methods are valuable; they are more fine-grained than expression and result accuracy.

Reasons to reject:
* The performance of the proposed approach on other datasets (such as Math, SVAMP, or ASDIV) is missing. The authords explain that the datasets they do use (Math23k and MAWPS) are used by many of the baselines. Experiments on MathQA and SVAMP will be added to the final version.
* Comparison of performance with LLMs is missing. The authors address this point in the rebuttal.
* Qualitative analysis of the unified tree structure is missing. The authors point to existing qualitiative analysis and will add further visualization.